# Opinions on the Orchard of the Lower Segura River: A Changing Space under Great Human Pressure

**María Dolores Ponce-Sánchez** [1], **Ramón García-Marín** [1], **Gregorio Canales-Martínez** [2] **and Javier Lozano-Parra** [3,*]

1   Departamento de Geografía, Universidad de Murcia, Campus La Merced, 30001 Murcia, Spain; maponce@um.es (M.D.P.-S.); ramongm@um.es (R.G.-M.)
2   Instituto Interuniversitario de Geografía, Universidad de Alicante, 03080 Alicante, Spain; gregorio.canales@ua.es
3   Instituto de Geografía, Pontificia Universidad Católica de Chile, Avda. Vicuña Mackenna, 4860 Santiago de Chile, Chile
*   Correspondence: jlozano@uc.cl

**Abstract:** What makes the Lower Segura River (southeast of Spain) unique is the existence of a densely populated and entropized territory around a smallholder agricultural activity that for centuries has shaped the so-called "Segura Orchard." In recent years, there has been widespread occupation due to the construction of secondary residences, which has clearly changed the rural appearance, sometimes creating an image more typical of disorderly residential urbanization than of an inhabited agricultural territory. The objective of this paper is to determine the attitude and position of the resident population regarding the situation and future prospects that are envisioned for this area. In this paper, we have conducted a personal and open interview as a technique to collect the information. If we are not capable of generating sustainable socio-economic activity, we will not be able to preserve, protect and transcend what we still know as an orchard. This spatial structure is undoubtedly singular and complex, and after a process of loss of identity and alteration of traditional uses, it requires an intervention that tends to organize it and protect the important territorial heritage that it still preserves.

**Keywords:** orchard; heritage; degradation; social perception; Southeast of Spain

## 1. Introduction

The present study shows the impressions stated by the inhabitants about the reality of this natural region, located in the Southeast of Spain. According to the Royal Academy of Spanish Language, the word "opinion" indicates to emit a judgment or assessment about something or someone; and it comes from the Vulgar Latin "parescĕre." So, this is qualitative research. One of the first scientific disciplines related to the study of perception was psychology ([1–4] among others). In general terms, psychology defines the concept of perception as "the cognitive process of consciousness that consists of the recognition, interpretation and significance for the elaboration of judgments about the sensations obtained from the physical and social environment, in which other psychic processes intervene (learning, memory and symbolisation)" [5].

The theoretical framework that supports this study starts from the concept of comprehensive sciences that the historian Dilthey exposed in the 19th century, taken up by various philosophers such as Husserl, Heidegger, Merleau-Ponty, the sociology of Weber, Súlchtz, Bordieu and the dialectic of Habermas, that argues that it is possible to understand and be critical, since reality is presented as a field of interest and the expository language that each person wields reflects their point of view, being

able to agree or contradict others [6]. The structuring bases of this approach are epistemologically based on (i) understanding and (ii) interpreting; through the first, reflection and contextualisation are exercised, the practice of which constitutes an act of hermeneutical knowledge that brings together the observer and the observed; on the second, in the process of analysis, the field material, its peculiarity and specificity, is valued, respectfully looking at the truth of the narrator that cannot be underestimated, but needs to be contrasted. These exercises require the following substantives: (i) experience (use that the narrator makes of the territory, shaped by culture and other multiple reasons; a person generates personal criteria resulting from what is learned and experienced socially); (ii) common sense (expressed as the accumulation of knowledge and the intellectual typifications generated collectively); (iii) action (behaviour or way of relating in a consensual way in society); (iv) significance (potentially interpreting what is perceived and developing a capacity to produce criticism); and finally, (v) intentionality (existential and non-rational behaviour in which consciousness intervenes).

Concern and commitment to environmental issues, the rational use of natural resources, as well as the search for a better quality of life, can be considered an ideology characteristic of today's society, and arises to denounce the serious injuries suffered by the territory. This circumstance is perceptible and alarming in the cultural landscapes of Mediterranean countries, created and maintained by primary activities, since they are being rapidly degraded ([7–11] among others). The economic development model has transmitted to the community some very effective status and comfort values for the interests of the power groups, who have promoted and allowed an urbanization process with dire consequences for the banks of the Segura river in general, and particularly in the lower section of its hydrographic basin, in the province of Alicante. This urban development, in addition to being unequal in terms of distribution of benefits, has led to the loss of large areas of cultivation in the traditional Orchard, considered today as a historical heritage of great value [12–15]. Currently, urban, industrial, tertiary, and infrastructure development (both communication and transport), among others, is engulfing the territory to the detriment of agricultural production space and the advancement of marginally disconnected areas or areas with uncompetitive agriculture. In recent years, natural vegetation (reed and other typical floodplain plants) has re-colonized the abandoned plots close to the coast or urban suburbs (Figure 1). Negative impacts limit the economic viability of the agricultural sector, while causing environmental degradation and the deterioration of social relations between the city and the countryside; a situation similar to that analysed for the Huerta de Valencia [16].

Over the centuries, an orchard space has been shaped, a human setting where the communities that have taken advantage of the alluvial plain of the river have built a social capital embodied in a unique heritage manifesting knowledge, beliefs, uses, rites, traditions, and customs that define the identity of its inhabitants. In short, a culture, which is what gives life to human beings and that brings together various dimensions and social functions by creating a way of living, social cohesion, creation of wealth and employment, as well as a territorial balance. In this sense, as scholars from rural areas affirm, "culture is alive" and whatever form it takes, it constitutes the best and most effective means of progress, because it contributes to the valuation of collective potential and favours the growth of the personality of individuals [17,18]. Thus, the protagonists interviewed in this study issue a qualitative assessment that reaffirms the uniqueness of the identity and culture they possess, as a result of the emotion caused by the modernisation of life experiences, what Ortega Gasset [19] called "experiences", a dynamic process that provides meaning to what is done, to what is undertaken, with a sustained character in time and space. This orchard space is, therefore, a secular agrarian territory, which reached the second half of the 20th century almost intact, maintaining its idiosyncrasy and cultural heritage from generation to generation, due to its economic viability and in the absence of other sources of wealth.

Given these circumstances of degradation of the Orchard by the appearance of other land uses much more profitable in the short term (residential construction for mainly foreigners and related services), the need to protect and revitalise an agricultural space of extraordinary cultural, natural, and economic value prevails. The Orchard must adapt to the new society and global world, but it

must also remain without losing its essence and the extraordinary opportunities it offers. To meet this objective, the maximum possible social consensus needs to be achieved.

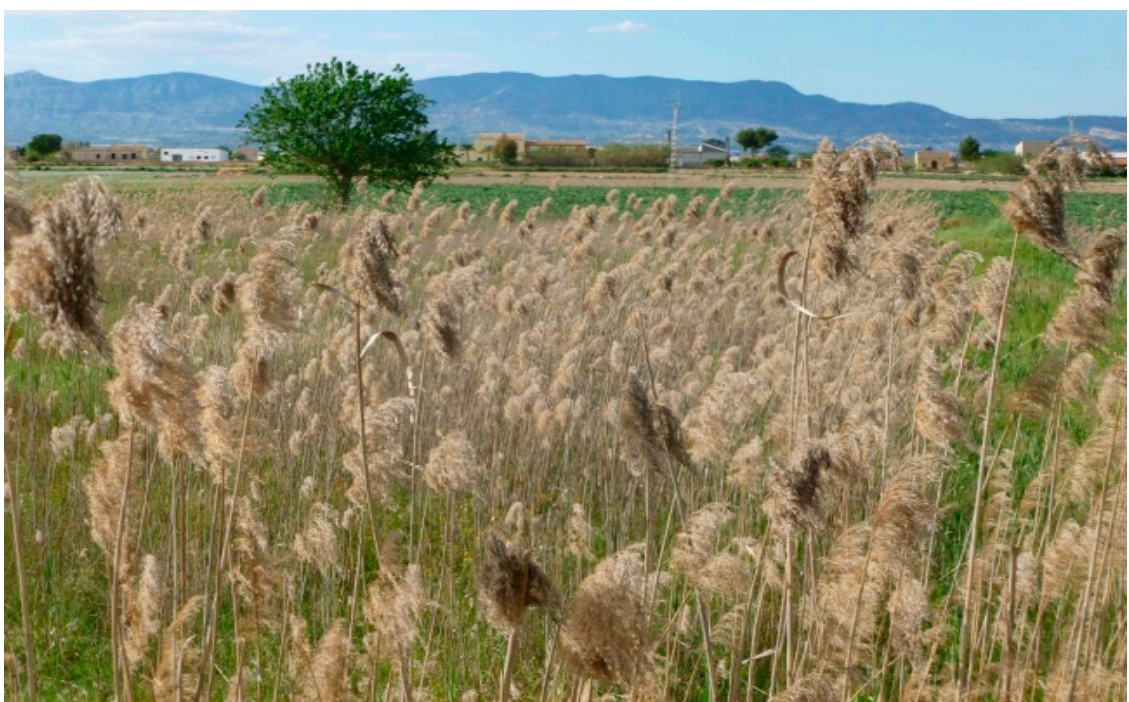

**Figure 1.** Abandoned cultivation plot in the municipality of Dolores (pending land use change), where the reed has recolonised the drained alluvial plain in the 18th century. Author: Gregorio Canales-Martínez.

The starting hypothesis of this research raises the existence of identity stories within society; the population chooses which aspects of their culture are worthy of appreciation and conservation and induce commitment to their daily spaces so that better use and exploitation of available resources is possible. Therefore, the objective of this paper is to determine the attitude and position of the resident population regarding the situation and future prospects that are envisioned for this area. Development projects must be carried out for everyone who inhabits the area, and for this reason it is legitimate for them to express what territory they want to improve their quality of life and well-being. The subjective space must be considered together with the objective, analysed and projected from the political sphere by technicians and authorities, sometimes unaware of spatial realities. After the perception analysis was carried out, we propose active measures to conserve and promote a symbolic and identifiable space for the resident population.

## 2. Study Area

The space that is analysed in this study, the Orchard of the Lower Segura river (Figure 2), was called "Huerta de Orihuela" (Orihuela Orchard) until the middle of the 20th century, and prevailed as an identifying reference of the geographical scope, in clear reference to the preeminence acquired by the name of the main city in the most immediate environment [20].

This territory currently represents one of the various intensive irrigated sectors that characterise the south of Alicante, and which has resulted in the appearance of numerous agricultural landscapes related to water. In essence, this landscape is considered the link between nature and culture, as it is the result of human action on the environment and constitutes an important heritage manifestation [21]. The current dominant irrigation system in this Orchard (whose terminology largely expresses its Muslim origin) has given rise to unique architecture in the distribution of water, which begins with the

reuse of the flow of the Segura river and the drainage generated in the Orchard of Murcia (upstream). There are numerous networks of irrigation ditches that start from weirs (Figure 3), and whose mission is to distribute the irrigation water by gravity, through a dense and hierarchical mesh of different sizes, which by flood spreads the water over cultivated lands. This broad supply infrastructure is duplicated in another of inverse characteristics, called the "azarbes" network (drainage ditches), whose function is to drain the soil (thus avoiding puddling),return the flows to the river by gravity, and to recover them again in the next dam, downstream of the previous one. In this way, the primitive colonisers achieved a well-organised use of the few available river contributions, at the same time that they achieved a complete reuse of water resources in the lower section of the Segura river. This complex irrigation system, with irrigation and drainage ditches for the use and collection of surpluses, gives rise to the double circulation of "living and dead waters" [22]. In this region, living water channels serve to irrigate farmland, and distribute the water derived from the river, while dead water channels serve to receive drainage from the lands, discharging them from the excessive humidity that damages them. This "dirty" water is reused for irrigation of downstream land.

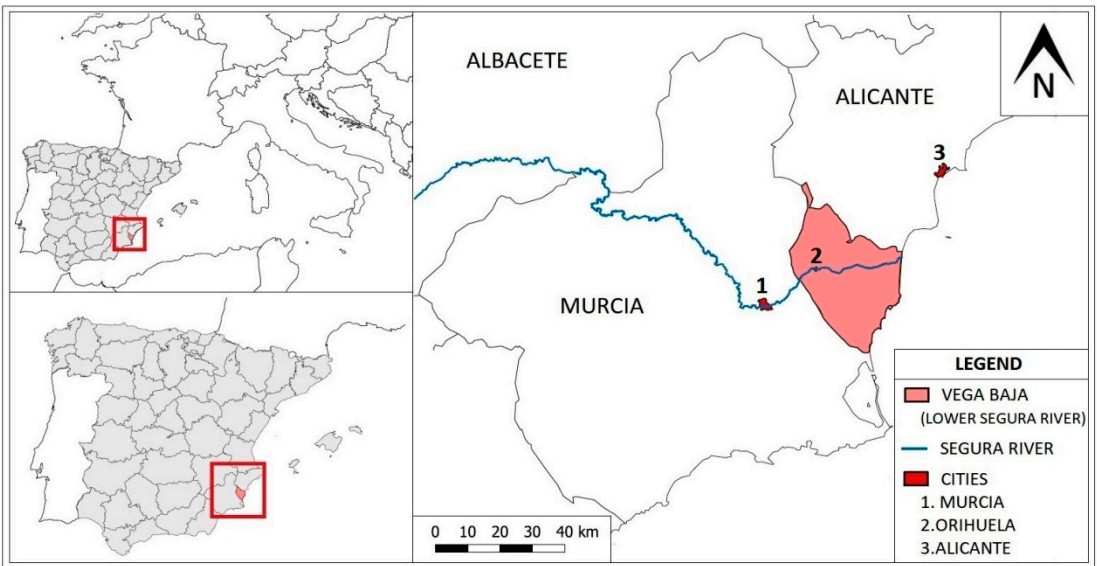

**Figure 2.** Location of the study area, and important nearby cities.

We highlight the high profitability of irrigated soil, since it makes it possible to harvest up to four crops a year on the same plot if planted with arable crops (Figure 3), due to the optimal fertility of a soil enriched with silt after each flood. Human occupation of the territory has conditioned a way of life linked to water, visible in a vast cultural legacy (both material and immaterial) that remains in force, although socially less and less recognised [23–26].

Currently, this production model, by virtue of the evolution that it has undergone in recent decades, is looked upon with negative connotations. Indeed, when touring the huertano (district) space, the observed perception is far from the traditional image; fruit of the introduction of other interests and expectations of an economic nature (real estate market) which have changed the value of the land (from production to consumption) due to the soil demands to accommodate new residents (Figure 4) who come to this territory due to the accelerated development of the tourism sector [27,28].

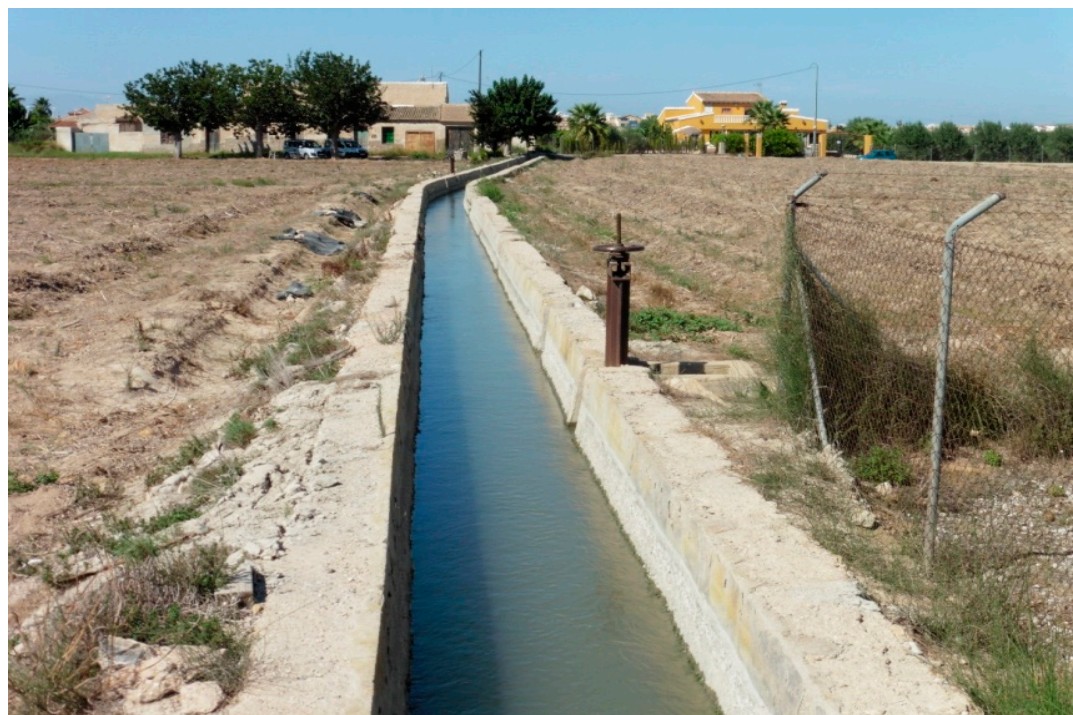

**Figure 3.** Main canal in Formentera village. Traditional urbanisation in the background, compared to the residential model implemented in recent years. Author: Gregorio Canales-Martínez.

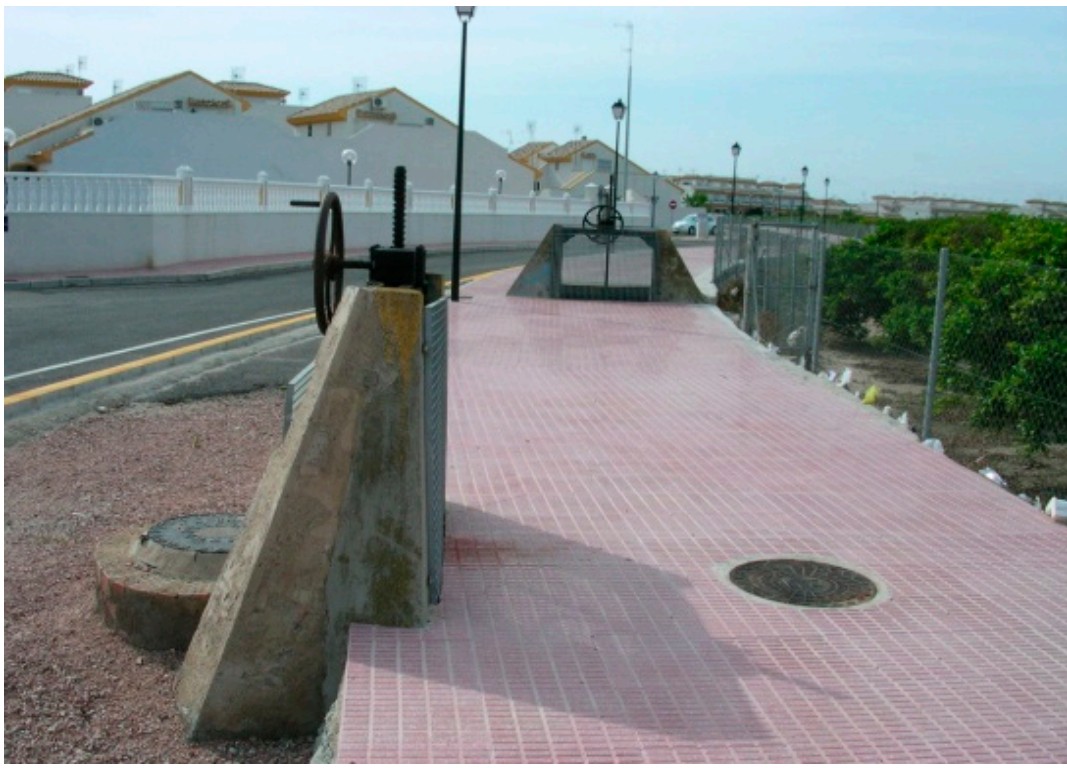

**Figure 4.** Main canal in Formentera village. The advance of residential use has motivated the concealment of the hydraulic infrastructure under the ground, essential for the survival of the Orchard. Author: Gregorio Canales-Martínez.

## 3. Methodology

### 3.1. Theoretical Background

The geography of perception or of representations makes the methodology that the present study follows possible, giving it an applied character. A current line of research, of extraordinary interest and wide scope for this discipline, although it started in the United States in the early sixties, coincides with geographers, urban planners and specialists in urban design and planning [29,30]. Lowenthal [31] drew attention to the need to carry out these studies, making a broad synthesis of work produced by psychology. He was the first to explore personal geographies; that is, the intimate vision of the world mixed with the imagination that each person possesses and the egocentric nature of the experience and that vision, as well as the influence of the sociocultural context on the formation of certain collective basic guidelines [32,33]. In short, a cognitive component intervenes in this process, which provides multiple points of view and habits, depending on the subjects, their cultures, ages, or sex. To explain them, it is necessary to resort to mental representations. From the latter, they can be understood as "any of the creations, collective or particular, extracted from reality through the perceptual mechanisms of the human being, especially that of sight: the retina retains what it chooses or stands out, an image or images that form a representation" [34]. We should not forget, paraphrasing Gould [35], that the landscapes on which Geography bases its analysis are nothing more than the spatial expression of people's decisions. In this sense, authors like Gumuchian [36] prefer to speak of "represented space" and not of "perceived space", since only the perceptual mechanisms (sight, smell, hearing, taste, and touch) intervene in the latter, not the cultural context or cognitive processes.

In this paper, we have chosen to involve the population directly affected by what is happening in the orchard area. We have conducted a personal and open interview as a technique to collect the information [37–39], which constitutes a contribution to the diagnosis prior to the creation of spatial action plans. This information is of great importance, since it reveals problems, dysfunctionalities, and even solutions that had not been considered from the administrative sphere. We propose, in this way, to make citizen participation not only a style of government, but an unavoidable instrument in decision-making processes or the design of policies and programs for spatial planning [40].

According to the Report on mechanisms and instruments of coordination for the implementation of the 2030 Agenda in Spain [41], the concept of governance is defined as the set of different processes and methods through which individuals and institutions, public and private, manage common affairs. On the other hand, and at European level, according to the White Paper of the Committee of the Regions on Multilevel Governance (2009) [42], public participation is essential, since "it favours a more dynamic approach and greater accountability of the different agents". In short, among the mechanisms to govern, collaboration with society and transparency must stand out.

### 3.2. Materials and Methods

The interview, a method of a qualitative nature, involves a meeting between the researcher and the informants, which makes it possible to collect additional and, sometimes, unforeseen information of great interest. A useful instrument that brings the researcher closer to a better understanding of the lived space. Kayser [43] points out his ambivalence, because the subjectivity of the interviewees produces objectivity, and that is when the scientist verifies and appreciates behaviours and strategies.

The structure and content of the questionnaire obeys the established objectives; we propose various questions that make up three main sections:

i.   The first one tries to discover the socioeconomic profile of those interviewed, variables such as age, sex, and profession can determine the responses issued.

ii.  With the second group of questions we try to find out the degree of relationship that they have with the Orchard, as well as whether they are aware of the current situation in which this agrarian space is found. We urge you to answer the following questions: Do you reside in the Orchard or urban nucleus? How much time do you dedicate to agricultural work? What does

the Orchard mean to you? What are the factors that have caused the important loss of this agrarian landscape?

iii.     Thirdly, we are interested in knowing the reasoned opinion of these residents on the need to recover the space of which they are part, so the questions raised are the following: Do you consider it necessary to protect this agrarian landscape? Do you think that abandoned agrarian spaces should be revitalised ecologically and economically? What are the factors that have caused the significant loss of this agrarian landscape?

The data, eminently qualitative, has been coded in a database to achieve the results, which in themselves constitute important conclusions.

## 4. Results and Discussion

### Perception of the Orchard Space and Stories about the Inhabited Space

The need to consider the subjective space in terms of spatial planning becomes an inevitable necessity in this territory, still absent in most decision-making processes in this regard.

### *4.1. Socio-Demographic Characteristics of the Interviewees*

In the perception analysis, it is convenient to consider certain variables that condition the responses: age, sex, place of residence, profession, or direct link with the irrigated land, in this specific case.

Of the eight hundred and four interviewees, half are young, aged between 18 to 35 (Figure 5). The under twenties stand out, a group of special interest for this study, since their opinions are very interesting to perceive the value that this territory has for these generations not related to agricultural work, but who nevertheless have a greater environmental awareness. The other large group of people interviewed are adults, among this group people of 46 to 50 and 51 to 55 stand out. Both add up to the same percentage as the youngest (which proves the balance of the sample). Their opinions are also very interesting, as they have lived through the flourishing stage of agricultural production in the Orchard. The results reveal that 56.7% are men and the rest are women (Figure 6).

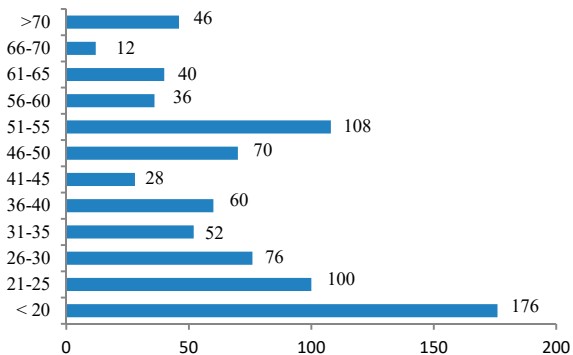

**Figure 5.** Interviews answered according to age intervals.

If we look at the professions represented, the diversity of economic activities is striking (Figure 7), reflecting the multifunctionality that characterises the study territory. The Orchard has historically been the economic sector par excellence, and agricultural activity dominates among those surveyed (19%), second only to students (30%). After them, a wide range of activities typical of urban areas is represented. Eighty percent of the sample resides in the main localities, and only the remaining 20% still do so in a scattered way in the Orchard of Lower Segura River.

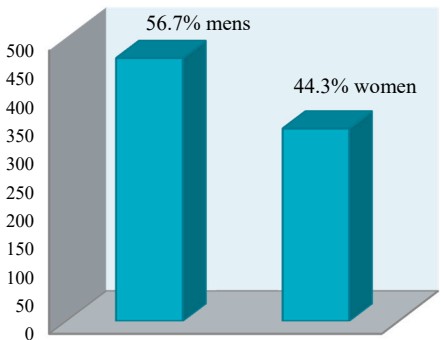

**Figure 6.** Interviewees by sex.

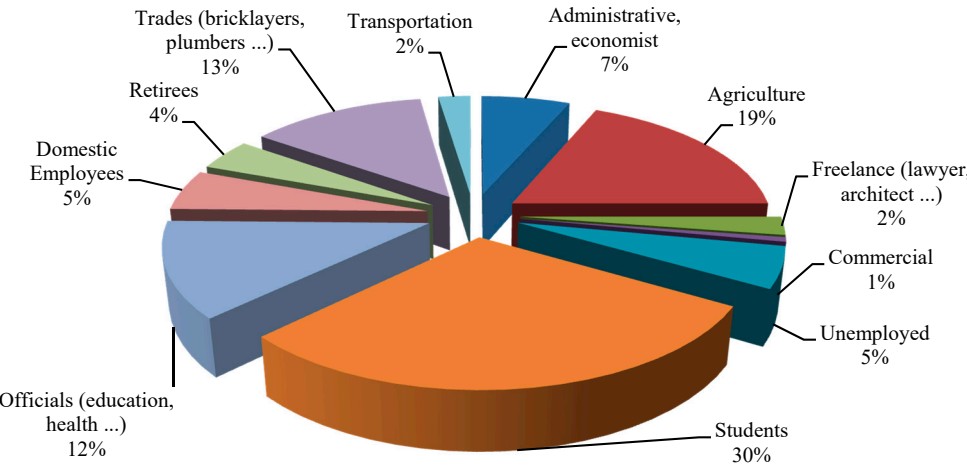

**Figure 7.** Interviewees's professional occupation.

*4.2. The Significance of the Orchard for the Interviewed Population*

In this section, the assessment given by the interviewees is highly significant. A very positive perception is surprising if one takes into account that fieldwork is carried out over the years in which agriculture is immersed in a period of crisis, as a result of the new economic orientation that is given to that space by the change of use of agricultural soil to urban. In most of the answers, the interviewees say that this agricultural space, by family tradition, is intimately linked to its existence. The emotional bond is a constant in almost all cases, as these irrigated lands constitute not only a fundamental source of income for those interviewed, as will be seen later, but also, the Orchard appears as an emotional territory with multiple beneficial qualities for the population. Very significant phrases are observed: "an open, natural, healthy and relaxing space" or "I disconnect from the world and I dedicate most of my free time to it". Only a few interviewees declare that they do not feel any affection for the Orchard space, although they recognise that its origin is foreign (Table 1).

**Table 1.** Perception of the meaning of the Orchard landscape.

| Ratings. | Number of Appointments | Percentage Value (%) |
|---|---|---|
| Production space | 492 | 34.4 |
| Improves quality of life | 352 | 24.5 |
| Culture, tradition, identity | 256 | 17.9 |
| Nature | 224 | 15.6 |
| Backspace, crisis | 100 | 7.0 |
| No affection, no link | 8 | 0.6 |
| Total | 1432 | 100.0 |

There is a conflict of interest between those who advocate seeking the profitability of agriculture (Figure 8) and those who propose the conversion of soil to a tourist-residential use. Although there are many interviewees who reflect a defeatist sentiment in the face of the current crisis in the primary sector, they recognise the importance that this intensive production space had until a few years ago: "currently, in my family, irrigated lands are a complement to our incomes derived from other trades, because depending on agriculture is complex and difficult".

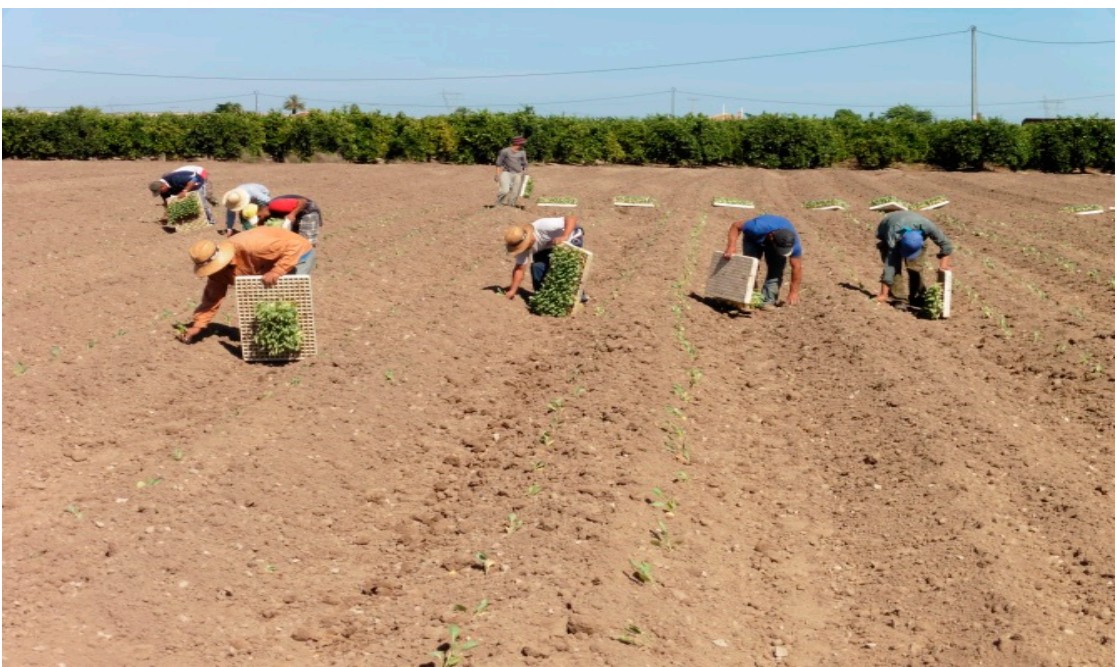

**Figure 8.** The smallholding, characteristic of this agrarian landscape, motivates manual work in the plantation of herbaceous crops (broccoli in this case). Author: Gregorio Canales-Martínez.

However, a longing for the past prevails, and the allusions about the traditional function that the Orchard represented are continuous: "It means a lot to me, since it is a unique resource in the Vega Baja (Lower Segura river) and the population should appreciate and value it"; "The Orchard has constituted the livelihood of my parents and ancestors, and it is a pity that now it has no place in society"; "It should not be lost, as it is part of our culture." The current situation of abandonment is also a consequence of the lack of profitability of a smallholding, which predominates in the land ownership structure, as well as the competition created by large agro-industrial companies. Some of the interviewees recognize that "if the current situation does not change, it is possible that I will continue to use my land for agricultural purposes, but only for my own consumption, and not for commercial purposes."

The productive nature of the Orchard is the most outstanding value in the responses collected. Truly, the vision given as a productive space is overwhelming, as for most of the interviewees it is "the main source of income" for this territory. Many other responses allude to the excellent quality of local fruit and vegetable products. In addition, we also verify that there is awareness among the interviewed population of the large amount of labour required by the different agricultural tasks (we must remember that up to four vegetable crops are harvested per year) and that a high percentage of work is carried out by women (collection, handling, packaging, etc.).

In the answers related to the significance of the Orchard, other closely related nuances prevail: the Orchard offers an excellent quality of life, considerable culture, and a space with "nature" (some answers: "A place to go and relax"; "An area where you can live peacefully, without the noise and pollution that there is in cities"). These perceptions are all behind the productive function, even though the increase in leisure time in post-industrial society has motivated a new re-evaluation of the orchard

environment as a place of recreation in contact with "nature", a role that has always been fulfilled by the surrounding populations and which we now explicitly record in the responses obtained.

The assimilation of the orchard and nature is constantly repeated throughout the interviews, a variable that appears together with the terms of landscape, tradition, culture, and heritage. Despite the productive character of these irrigated lands, the idea prevails that this activity is respectful to the environment ("The Orchard is a means of production, but it must be considered an ecological corridor that improves environmental quality, and therefore it must be preserved as a social and community resource").

### 4.3. Factors Causing the Loss of Orchard Space

In this section, the responses obtained show a variety of circumstances that have continuously diminished the presence of the Orchard, both in the territory and in the imagination of the population. In total, eight variables have been collected to show the reality that this productive space is going through (Table 2).

**Table 2.** Perception of the factors that have caused the loss of the Orchard space.

| Ratings. | Number of Appointments | Percentage Value (%) |
| --- | --- | --- |
| Speculation | 388 | 24.4 |
| Change of mentality (Rural to Urban) | 324 | 19.5 |
| Low profitability | 260 | 16.4 |
| Attractiveness to other production sectors | 188 | 11.9 |
| Inadequate policies | 140 | 8.8 |
| Lack of water resources | 108 | 6.8 |
| Disappearance of family farming | 92 | 6.8 |
| Lack of spatial planning | 84 | 5.3 |
| Total | 1584 | 100.0 |

i.　　Speculation. The accelerated economic change experienced in recent decades has triggered, in a traditionally agricultural region, a new production model that has evolved towards the search for immediate profits, generating a radical change in land use. The interviewees indicate speculation as the main factor causing the loss of the Orchard, and they refer emphatically to the real estate sector as an activity that reduces the irrigated area (Figure 9). Some examples of responses are as follows: "The Orchard gives way to urban speculation, which increases the value of the land and generates rapid and excessive enrichment for construction companies", "Many of the agricultural workers traded their jobs for construction during the housing boom". Housing construction has been the economic model that has focused regional development in recent decades [44].

ii.　　Change of mentality. The predominance of urban population and ways of life associated with the city is another cause stated by interviewees. The city is associated with well-being, the opposite of all the "discomforts of living in rural areas." This perception is especially prevalent in the young population surveyed, a circumstance that motivates the elderly to exclaim the lack of generational replacement in the work of the orchard (Example response: "Today we can find everything in the city, but if we live in the orchard, we are socially isolated and we need to take our car to access the stores, banks, health centres...").

iii.　　Low profitability. In almost all the responses, the incidence of the economic factor is strongly apparent. Some of the most expressive answers are as follows: "The agricultural lands are abandoned due to the low profitability and the high competitiveness of the produce that comes from abroad"; "The small and medium farmer cannot subsist with the current market prices"; "Imported produce has a lower cost than what we grow here, and we do not get adequate profitability"; "Today you cannot live off agricultural activity, it is no longer profitable, you have to abandon this activity". In the interviews there is a feeling of discouragement and great

sadness. The traditional individualism of the farmer and the mistrust towards other marketing systems, where associations acquire a prominent role, are barely consolidated in the Orchard of the Lower Segura River.

iv. Attractiveness in other production sectors. Three valuations in which these changes are observed are described below, and are reflected consistently with the age of each person surveyed: (a) A 79-year-old pensioner points out the following: "There is a lack of interest from the new generations in agricultural jobs, with unpredictable hours, young people are in favour of the advantage of an eight-hour work day and having free time"; (b) A 50-year-old person said: "There is a loss of attractiveness for young people and adults who inherit the farms, and due to the lack of amenities offered by the rural environment they choose the urban space, where they get adequate wages for the effort made, and with older labour and cultural possibilities"; (c) A 26-year-old young woman conveyed the popular belief that "people who work in the agrarian sector are clumsy, uncivilized, and poorly educated."

v. Inadequate policies. Direct criticism of politicians and the administration itself take on special relevance among those interviewed, whom they hold responsible for "the construction and speculative activities accompanied by permissive legislation", which has gone against the maintenance of agriculture and the consequent conservation of the Orchard. Local authorities have favoured agricultural land use ratings over developable land, a generator of higher capital gains and taxes. Some interviewees strongly and concisely point out that the main factor in the deterioration of the orchard space is related solely to "the attitude of the rulers" and "the unprofessionalism of politicians", sometimes allied with "urban speculators who have dedicated their time and effort to actions of high, immediate, and safe profitability".

vi. Lack of spatial planning. In the same sense, the surveys have seen that: "the absence of planning (in the medium and long term) and protection by the public powers or legislators" are a cause of the degradation of agriculture.

vii. Disappearance of family farming. The economic change that has occurred since the second half of the 20th century, with advances in new forms of communication and commercialisation, was a shock both to the mentality of people linked to agricultural land and for the deeply rooted social behaviours that had been practiced and transmitted from generation to generation. Smallholder agriculture, together with an elderly or retired active population linked to agricultural work, with the aforementioned problems, has caused the disappearance of small family farms. Respondents also attribute this situation to the appearance of large fruit and vegetable companies, which, in the opinion of many interviewees: "devour small farmers", who, in the absence of capitalisation and renewal of production systems, are forced to "lease the farmland for large agribusinesses".

viii. Lack of water resources. The scarcity of water resources and the poor quality of the flows used in irrigation are also a point of contention when discussing the situation that this agricultural area is going through. Indeed, to the structural water deficit characteristic of the Segura river, we must add the location of the sector that is analysed in the final section (near its mouth), where the reduced volumes that arrive cause the so-called "water wars", due to the confrontation that there is among irrigation farmers. We also warn that farmers are unhappy with the level of contamination of the river waters in this last stretch or irrigated section: "the orchards cannot withstand water of such poor quality!"

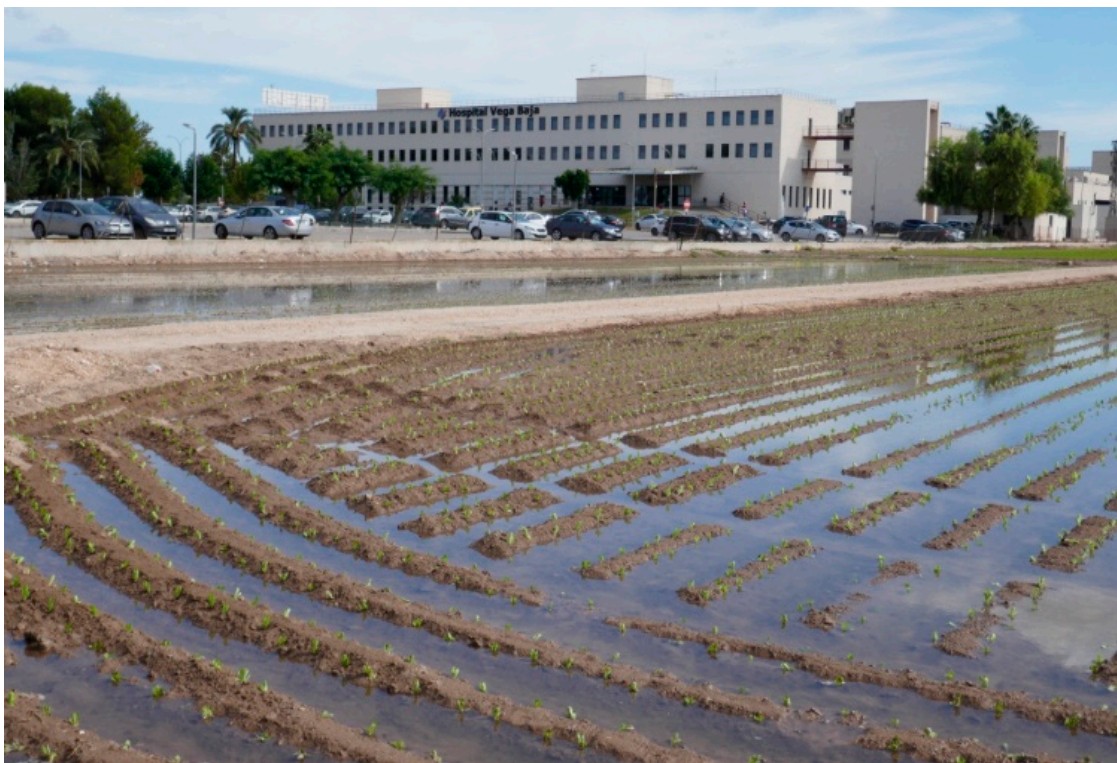

**Figure 9.** Public hospital located on the flood bed of the Segura river. The growth of service infrastructures for the resident population also reduces the cultivated space. Author: Gregorio Canales-Martínez.

### 4.4. Considerations Regarding the Protection of the Orchard

In this section, the responses of the interviewees to the following questions are analysed: Would you protect the existing Orchard? Would you revitalise, ecologically and economically, the abandoned or deteriorated agricultural spaces? The total statements collected have been grouped into seven major variables (Table 3).

**Table 3.** Perception of the need to protect the Orchard.

| Ratings. | Number of Appointments | Percentage Value (%) |
|---|---|---|
| Improves economic diversification | 360 | 25.2 |
| Ensures food production | 276 | 19.3 |
| It constitutes a singular ecosystem | 220 | 15.4 |
| Essence of their lives | 192 | 13.4 |
| Maintain culture and identity | 144 | 10.1 |
| Landscape value | 120 | 8.4 |
| Excellent heritage | 116 | 8.1 |
| Total | 1428 | 100.0 |

In a first approximation, two outstanding perceptions related to fields as diverse as economic and cultural are to be reviewed, the first being well above the second, which shows, a priori, the importance that this productive sector still has for an important volume of the population that makes its livelihood from the Orchard. Furthermore, we demonstrate the high regard that respondents have of this space as a hallmark, since the agrarian landscape is part of its idiosyncrasy, and they associate it with the natural value of the territory. However, it is convenient to specify some of the most significant assessments registered in the field work.

In relation to the responses that indicate the economic importance of the Huerta, all of them coincide in pointing out that this function has been the dominant one for centuries: "the economy in our towns has been based on agriculture, which has satisfied the food needs of the entire population." This duality between the dominant economic sector and the guarantee of local supply is a constant among those surveyed. That is why many respondents are in favour of "recovering agriculture in the area, as it supplied raw materials to industry, local markets, and even had surpluses for export."

Along with the economic considerations, the interviewees see the Orchard as an outstanding "ecological space", even when it is recognised that this territory is the result of a long historical process of human creation. This value being the priority to give it legal protection. The idea of union that identifies the natural landscape with the cultural one is transmitted. In this sense, the symbiosis that exists between the farmer and the ecologist is not strange: the farmer takes care of "an environmental resource" increasingly appreciated among the parameters of quality of life and well-being of the built nuclei that surround the agricultural space. Many interviewees express as an alternative "an ecological orchard" and demand the intervention of public powers, aware that "in this way it would improve the economy of farmers". This feeling emerges in expressions such as: "we must protect our natural environment", or the Orchard "is a natural space that must be cared for, the work of the farmers being necessary". Likewise, the respondents recognise that this orchard landscape "constitutes an important historical and cultural heritage", accepted and seen by all as a "symbol of our identity."

A statement that jointly expresses all these values is the following: "The Orchard is part of the culture and history of this area and its progressive loss means a great loss of the common heritage. Conserving the Orchard entails multiplying the value and the ecological and economic possibility of the towns of the lower Segura River."

## 5. Conclusions

The Orchard of the Lower Segura river stands out as one of the cultural agrarian landscapes of the Mediterranean basin most affected by urban development, which implements various land uses (industrial, tertiary...) and causes an alarming decline of this millennial productive space.

From the analysis of all of the above, it appears that the Orchard constitutes the rural environment par excellence in this territory. It is noteworthy in this final balance of opinion, in which several generations have participated, to show how the appreciation of the younger population coincides with the opinions of adults, although the younger ones glimpse other perspectives more in line with today's society, without entering into contradiction with the traditional uses of this agricultural space. Thus, these perceptions can be specified in the following concepts and final considerations:

- Rediscovery. The high degree of awareness of the respondents in relation to the intrinsic values of the rural environment in which they live is striking. The sensitivity towards the protection and conservation of the surrounding rural space stands out greatly, giving special meaning to the landscape, as it constitutes a cultural benchmark characterised by its uniqueness.
- Attractive. The Orchard implies a suggestive heritage area for its population, derived from the symbiosis of the varied cultural wealth in relation to the natural wealth that surrounds it.
- Farming. The agricultural value underlies the thoughts of the residents when it comes to evoking the term Orchard, a word closely linked to quality products harvested in the vicinity, which means travelling a short distance to reach local markets in optimal conditions of freshness and avoiding the increase in prices, by reducing the costs derived from transport.
- Multiactivity. The results obtained from the interviews demonstrate the importance that polyfunctionality can be achieved in this agricultural space, understood in a rational and balanced way. The implantation of other activities derived from leisure society, be they recreation for the surrounding populations, or tourism for a foreign clientele, contribute to energising the socioeconomic productive fabric. In this sense, agrotourism could be the guarantor of the maintenance of the landscape and the built heritage.

The strategies to be promoted to improve the current situation of the "Orihuela Orchard", according to the surveys, involve developing the following aspects:

- We recommend improving the existing production and commercial system to achieve greater agricultural development. In an agrarian space such as the one analysed, which has in its favour a solid image in the collective sentiment, linked to the agricultural landscape, farmers should join forces (together with the intervention of local political authorities) to incorporate into their production figures (brands) that conferred recognised quality, such as: Protected Designations of Origin, Protected Geographical Indicators, or Guaranteed Traditional Specialty certifications. With these measures it would be possible to increase competitiveness and position production favourably in the national and international market.
- We consider it necessary to obtain, for this irrigated orchard territory, the legal construct of "Agrarian Park", and although this legal form is not included in the regulations of the Valencian Community, we know that it has given positive results in other Spanish and European areas. The implementation of this legal protection figure would mean considering this area of historical irrigation as a common good, and preserving it from urban progress.
- Lastly, it would be necessary, as stated in the Baeza Charter on Agrarian Heritage [45], to achieve the dignity of agrarian activity to avoid losing an important part of the knowledge and culture that society has developed in rural spaces. For this, it is necessary to understand agrarian heritage holistically, as a broad set of natural and cultural assets, integrating all possible manifestations: technological, documentary, ritual, and linguistic, among others.

Orihuela Orchard, like other areas of peri-urban agriculture, requires that those responsible for spatial planning bet on the recommendations of the Technical Decision on "Peri-urban agriculture" of the European Economic and Social Committee (2005) that urges to reach: (i) A territorial proposal ensured by regulations dedicated to protecting and promoting agrarian activity and the land market for agriculture; (ii) Instruments and means aimed at guaranteeing the stability of agricultural land to eradicate the future urbanization threat; and (iii) A comprehensive management, administered by an entity dedicated to disseminating knowledge about the multiple values of these agricultural territories, and committed to sustainable development, with the agreement of all the agents involved: citizens, public administrations, and farmers.

The University of Alicante, under the direction of Professor Gregorio Canales, and since 2006, promotes courses and workshops for the preservation, defence, and dissemination of the values of these peri-urban orchards. Concrete actions are taken with the participation of farmers, irrigation managers, various county associations, and policy makers. The contributions of these events have been reflected in concise manifests that demand, in addition to the proposals of the European technical decision mentioned above, pressing interventions such as: (i) promotion of organic farming, backed by a quality seal or designation of origin, to promote better competitiveness in international markets and ensure profitability of farms; (ii) improve and modernize the local production system, without altering its good work in reuse of water flows, and articulate this function with agribusiness, favouring professional qualification; (iii) create land banks with parcels that have been left uncultivated, and thus maintain the unique biodiversity associated to this peri-urban agriculture; (iv) generate decent income for owners and tenants, and boost agricultural employment; and (v) catalogue and protect the traditional irrigation system (weirs, ditches, and other hydraulic heritage), to allow its ancestral operation and its value as structuring elements of Orchard landscape. All these instructions are not possible without a sincere will on the part of the administrations closest to the population (councils and government of autonomous community). However, a general social requirement along with that of farmers would help enormously.

**Author Contributions:** M.D.P.-S., R.G.-M., and G.C.-M. conceived the idea, wrote and reviewed the original draft in the same way. J.L.-P. edited it and contributed by the funding acquisition. All authors have read and agreed to the published version of the manuscript.

**Funding:** This research was funded by the Fondo Nacional de Desarrollo Científico y Tecnológico (FONDECYT 11161097) granted by the Government of Chile. Authors also thanks to Instituto de Geografía of Pontificia Universidad Católica de Chile for its economic support. Finally, authors thanks to anonymous reviewers for their suggestions which have undoubtedly improved this study.

**Acknowledgments:** 

**Conflicts of Interest:** The authors declare no conflicts of interest. The funders had no role in the design of the study; in the collection, analyses, or interpretation of data; in the writing of the manuscript, or in the decision to publish the results.

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
