# Peer review of "Opinions on the Orchard of the Lower Segura River: A Changing Space under Great Human Pressure"

_sustainability, doi:10.3390/su12114446_

Round 1

Reviewer 1 Report

The paper is well structured and only the following revisions and suggestions are recommended:

- Table 1. It is suggested to order the items of the first column with reference to the values (in ascending order) of the second and third column (i.e. production space, improves quality of life, etc.). In this way the results are more readable. It is also possible to insert a one or two graphs that help to read the table or directly to present the results.

- Table 2 and Table 3. As in Table 1 I suggest to order the items of the first column with reference to the values (in ascending order) of the second and third column, and/or insert one/tow graphics to present the results.

Author Response

Dear reviewer,

We greatly appreciate your observations and suggestions in our study. We think it has improved significantly after including them.

The revisions have been included in the uploaded documment and can be appreciated by the red font and by the track change comments.

We hope that our revisions be well received.

Kind regards.

Reviewer 2 Report

A significant research problem was undertaken, both in theoretical and practical aspects. The research is a part of the current popular research trend regarding the role of nature and culture heritage in the development perspectives of the modern economy and society. The benefit of the scientific rank of the article would be to enrich literature studies.

In view of the limited use of mathematical and statistical methods, the content of the article is mainly a research report and presentation of Author's views.

The Conclusion is very general. It seems that specific actions could be proposed to support the development of analyzed area, indicating the institutons and time perspective in which these actions could be implemented.

The whole is, however, an interesting research study that can serve as a source for conducting comperative research in other regions.

Author Response

(The authors gave the same response as above.)

Reviewer 3 Report

It was a pleasure to read the paper titled "Opinions on the Orchard of the Lower Segura River: a changing space under great human pressure"

lines 171 - 205 looks like theoretical background not methods. I suggest to create new chapter 

To improve readability I suggest to sort data in tables and fig. 4 from top score to the lowest. 

The paper shows the case study and conclusions refers to the Orchard area which is good. Sustainability is an international journal, so some general conclusions should be added, which could be applied to other similar areas.

Good luck with your paper

Author Response

(The authors gave the same response as above.)
